# Comparison of In-Hospital Major Adverse Cardiovascular Events in Patients with Acute Myocardial Infarction Treated with Ticagrelor or Clopidogrel in the Emergency Department: A Propensity Score Matched Retrospective Cohort Study

**DOI:** 10.3390/healthcare11162246

**Published:** 2023-08-10

**Authors:** Po-Yao Huang, Hong-Mo Shih, Szu-Wei Huang, Yan-Cheng Pan, Fen-Wei Huang, Wei-Kung Chen, Shao-Hua Yu

**Affiliations:** 1Department of Emergency Medicine, China Medical University Hospital, Taichung 40402, Taiwan; 2School of Medicine, College of Medicine, China Medical University, Taichung 40402, Taiwan; 3Department of Public Health, China Medical University, Taichung 40402, Taiwan

**Keywords:** dual antiplatelet therapy, acute myocardial infarction, clopidogrel, ticagrelor, major adverse cardiovascular event, emergency department

## Abstract

Background: Dual antiplatelet therapy (DAPT) is a standard treatment option for acute myocardial infarction (AMI). The difference between the efficacy of ticagrelor and clopidogrel in the emergency department (ED) before percutaneous coronary intervention (PCI) remains unknown. The present study compared the in-hospital major adverse cardiovascular event (MACE) rates between patients with AMI treated with clopidogrel and those treated with ticagrelor in the ED before PCI. Methods: We retrospectively collected the data of patients diagnosed as having AMI in the ED. Patients were only included if they had successfully received complete DAPT with aspirin and ticagrelor/clopidogrel in the ED and had undergone PCI. The patients were divided into two groups according to their DAPT regimen. The primary outcome was the rate of in-hospital MACEs. The secondary outcomes included an unexpected return to the ED within 72 h, readmission within 14 d, and revascularization. Results: A total of 1836 patients were enrolled. Patients in the ticagrelor group had a lower in-hospital MACE rate (3.01% versus 7.51%, *p* < 0.001) and in-hospital mortality rate (2.15% versus 5.70%, *p* < 0.001) than those in the clopidogrel group. Multivariate logistic regression analysis revealed ticagrelor was independently associated with a lower risk of in-hospital MACEs (odds ratio [OR]: 0.53, 95% CI: 0.32–0.88, *p* = 0.013). After propensity score matching, the risk of in-hospital MACEs remained significantly lower in the ticagrelor group (OR 0.42, 95% CI: 0.21–0.85, *p* = 0.016). Conclusion: DAPT with ticagrelor and aspirin in the ED before PCI is associated with a lower in-hospital MACE rate among patients with AMI.

## 1. Introduction

Acute myocardial infarction (AMI) is a true emergency disease necessitating emergency department (ED) visits. According to statistics from the Ministry of Health and Welfare in Taiwan, heart disease was the second most common cause of death in people older than 45 years in 2020. In Taiwan, 16,125 people were diagnosed as having AMI in the ED in 2018. In total, 18% of patients with acute coronary syndrome (ACS) have been reported to experience recurrent major adverse cardiovascular events (MACEs) [1]. Dual antiplatelet therapy (DAPT) plays a vital role in the treatment of AMI and has been proven to improve the prognosis of patients with AMI.

Aspirin reduces the frequency of ischemic complications after percutaneous coronary intervention (PCI) and should be administered in the periprocedural period. P2Y_12_ inhibitors are also essential for patients undergoing PCI. Previous clinical trials revealed that clopidogrel had similar efficacy to ticlopidine but lower rates of drug discontinuation attributable to noncardiac events [2]. The risk of ischemic events, including death, MI, and stroke, was found to decrease with the administration of a loading dose of clopidogrel and continuation of treatment up to 9 months after elective PCI [3]. However, clopidogrel is a prodrug that is transformed in two steps. After oral administration, clopidogrel is absorbed in the intestine and activated in the liver. The first step leads to the formation of 2-oxoclopidogrel, while the second step leads to the conversion of 2-oxoclopidogrel to the active metabolite. Some patients who have impaired hepatic cytochrome P450 enzymes responsible for clopidogrel metabolism may exhibit a diminished platelet response to clopidogrel [4,5,6].

Ticagrelor is another P2Y12 inhibitor commonly used in the ED. In the Platelet Inhibition and Patient Outcomes (PLATO) study, compared with clopidogrel, treatment with ticagrelor reduced the incidence rate of the composite endpoint of death from vascular causes, MI, or stroke [7]. Ticagrelor was also found to be associated with a lower rate of stent thrombosis. Although no significant differences were noted in the rates of study-defined bleeding events following the administration of ticagrelor, the incidence of noncoronary artery bypass graft (non-CABG) major bleeding was significantly higher among patients treated with ticagrelor than among those treated with clopidogrel. The Ticagrelor Versus Clopidogrel in Patients with STEMI Treated With Fibrinolysis (TREAT) trial revealed that ticagrelor was comparable to clopidogrel in terms of the rates of thrombolysis in myocardial infarction (TIMI) major bleeding, fatal bleeding, and intracranial bleeding in patients receiving fibrinolytic therapy for ST-elevation myocardial infarction (STEMI) [8]. Given the increased bleeding risk, ticagrelor should be used with caution in older patients. One study suggested that clopidogrel is a reasonable alternative for older patients with ACS undergoing PCI, with similar rates of ischemic events and less bleeding being reported [9].

The use of DAPT after PCI prevents stent thrombosis and reduces ischemic events at the cost of increased bleeding [10]. Loading of dual antiplatelet agents in the ED before PCI is also important to prevent MACEs during hospitalization [11]. However, few studies have compared the effect of loading of clopidogrel versus ticagrelor in preventing in-hospital MACEs in the ED. Therefore, this study compared in-hospital MACEs between patients with ACS treated with clopidogrel and those treated with ticagrelor in the ED before PCI.

## 2. Materials and Methods

### 2.1. Study Design and Population

This retrospective study included the data of patients with AMI who were admitted to the ED of China Medical University Hospital (CMUH), Taichung, Taiwan, from January 2016 to December 2019. CMUH is a tertiary center in Taiwan and has a monthly ED capacity of 14,000 patients. Approximately 650 patients with AMI visit this hospital annually. Moreover, approximately 550 coronary catheterization and 450 PCI procedures are performed at this hospital annually.

Patients were diagnosed as having AMI on the basis of ACS symptoms, elevated cardiac enzyme levels, and electrocardiography patterns. Patients who were diagnosed as having STEMI and non-STEMI (NSTEMI) and who underwent PCI were included.

We collected the data of patients with AMI in the ED. Patients were included only if they had successfully undergone PCI. Primary PCI was performed in STEMI patients in our study. Early PCI and selected PCI were performed in NSTEMI and UA patients. Patients without PCI or those diagnosed as having unstable angina were excluded. Patients who did not complete DAPT in the ED for any reason, who did not undergo coronary angiography, or who underwent coronary angiography without significant coronary artery stenosis were also excluded. Only patients with AMI who had significant coronary artery stenosis on angiography that necessitated PCI, including stenting or balloon dilatation, were enrolled in the study.

### 2.2. Data Collection

The timing of initiation of DAPT and the choice of P2Y_12_ inhibitors (clopidogrel or ticagrelor) were recorded for each patient in our study. The initiation of DAPT was defined as the first administration of aspirin and a P2Y_12_ inhibitor (clopidogrel, prasugrel, or ticagrelor) in the ED at a recommended loading dose. The loading dose of aspirin was 300 mg, clopidogrel 300 mg, and ticagrelor 180 mg, respectively. Different DAPT regimens (aspirin with clopidogrel or aspirin with ticagrelor) were prescribed to different patients. The patients with previous coronary artery disease who had an already administered maintenance dose of dual antiplatelet drugs were categorized in a subgroup of either group based on the original therapy (clopidogrel or ticagrelor). The patients’ demographic data, including age; sex; smoking history; and underlying diseases, such as hypertension, diabetes mellitus, coronary artery disease, cerebrovascular disease, chronic kidney disease, and hyperlipidemia, were also recorded. Moreover, the patients’ clinical features at presentation to the ED, including blood pressure; heart rate; peak troponin I level in the ED; renal functions, such as glomerular filtration rate and creatinine level; Killip scores for congestive heart failure; and left ventricular ejection fraction (LVEF), were recorded. The angiographic findings of each patient were also recorded.

This study was approved by the Institutional Review Board of China Medical University. Written informed consent was not obtained from the patients because of the retrospective nature of this study.

### 2.3. Outcome Measurement

The primary outcome were in-hospital MACEs, which was defined as all-cause mortality, ischemic stroke, recurrent MI, and unplanned repeat PCI. The secondary outcomes included an unexpected return to the ED within 72 h, readmission within 14 d, PCI performed within 30 d, or CABG surgery performed within 30 d after discharge from the hospital.

### 2.4. Statistical Analysis

The patients were divided into 2 groups. The clopidogrel group received aspirin and clopidogrel in the ED before PCI, whereas the ticagrelor group received aspirin and ticagrelor in the ED before PCI. Bivariate analysis was performed to assess the difference in patient characteristics, preexisting comorbidities, and clinical features at presentation to the ED between the 2 groups. Categorical variables were assessed using chi-squared tests and are presented as percentages. Continuous variables were assessed using independent-samples *t*-tests and are presented as the means ± standard deviations.

Furthermore, univariate analysis was performed to identify the factors associated with MACEs. Significant variables (*p* < 0.2) were included in a stepwise backward multivariate logistic regression analysis. To avoid selection bias between the two study groups, we performed propensity score matched (PSM) analysis. Factors that might influence the choice of ticagrelor or clopidogrel, including age, sex, prior coronary artery disease, prior cerebrovascular disease, chronic kidney disease, Killip classification, and STEMI or NSTEMI, were selected for PSM. PSM was performed using the nearest neighbor method, with a caliper width set at 0.1. After PSM, the standardized mean difference was utilized to assess the balance between the clopidogrel and ticagrelor groups, and the C-statistic was calculated to evaluate the predictive performance of the propensity score model. Subsequently, a conditional logistic regression analysis was conducted, adjusting for the same covariates, to identify the effects of ticagrelor and clopidogrel on the reduction in in-hospital MACEs. All statistical analyses were performed using SAS software version 9.4 (SAS Institute Inc., Cary, NC, USA). A two-tailed *p*-value of <0.05 was considered significant.

## 3. Results

A total of 1836 patients diagnosed as having AMI from January 2016 to December 2019 were included in this study. Of these, 135 patients who had incomplete data, received incomplete DAPT, were lost to follow-up, or did not undergo PCI were excluded. Of the 1836 patients, 938 patients were diagnosed as having NSTEMI and 898 were diagnosed as having STEMI. Moreover, 929 patients were included in the ticagrelor group, whereas 772 patients were included in the clopidogrel group (Figure 1).

The patients’ baseline characteristics are presented in Table 1. Patients in the clopidogrel group were older than those in the ticagrelor group (65.63 ± 13.49 versus 59.37 ± 13.08 years, *p* < 0.001). Although men were predominant in the two groups, the percentage of men was higher in the ticagrelor group (82.24% versus 71.89%, *p* < 0.001). Patients in the clopidogrel group had a higher prevalence of underlying diseases, such as hypertension (62.56% versus 54.79%, *p* = 0.001), diabetes mellitus (43.65% versus 31.75%, *p* < 0.001), coronary artery disease (29.53% versus 16.79%, *p* < 0.001), cerebrovascular events (7.90% versus 2.91%, *p* < 0.001), and chronic kidney disease (22.54% versus 9.36%, *p* < 0.001). Patients in the ticagrelor group had a higher prevalence of smoking (57.80% versus 47.80%, *p* < 0.001) and hyperlipidemia (9.15% versus 5.31%, *p* = 0.002).

Assessment of the clinical features at presentation revealed that the clopidogrel group had a higher Killip score (II to IV) (23.58% versus 15.61%, *p* < 0.001), higher peak troponin I level (5.19 ± 11.53 ng/mL versus 3.60 ± 9.36 ng/mL, *p* = 0.002), and higher creatinine level (2.64 ± 3.10 mg/dL versus 1.67 ± 3.88 mg/dL, *p* < 0.001).

The ticagrelor group had more patients with STEMI than the clopidogrel group (58.45% versus 37.82%, *p* < 0.001). By contrast, the clopidogrel group had more patients with NSTEMI than the ticagrelor group (41.55% versus 62.18%, *p* < 0.001). No significant difference was noted in the LVEF or significant coronary lesions (defined as >50% stenosis of the left main coronary artery or >75% stenosis of other coronary arteries) between the 2 groups (Table 1).

Patients in the ticagrelor group had better outcomes in terms of the in-hospital MACE rate than those in the clopidogrel group (3.01% versus 7.51%, *p* < 0.001). Moreover, the in-hospital mortality rate was lower in the ticagrelor group (2.15% versus 5.70%, *p* < 0.001). No significant difference was noted in the incidence of in-hospital stroke or recurrent MI between the 2 groups. The rate of PCI procedures performed within 30 days was higher in the ticagrelor group (17.22% versus 13.73%, *p* = 0.048); however, no significant difference was noted in other secondary outcomes, such as return to the ED within 72 h, readmission within 14 days, or CABG surgery performed within 30 d after discharge (Table 2).

Age, clopidogrel use, hypertension, diabetes mellitus, cerebrovascular disease, higher Killip scores, higher peak troponin I levels, left main coronary artery disease, and STEMI were associated with increased MACE rates in our univariate analysis. However, ticagrelor use, smoking, and high blood pressure were associated with decreased MACE rates. Variables that were significantly associated with MACEs in univariate logistic regression analysis and were reported as important prognostic factors in previous studies, [11,12] namely, age, sex, ticagrelor or clopidogrel use, smoking, hypertension, diabetes mellitus, cerebrovascular disease, Killip score, peak troponin I level, blood pressure, left main coronary artery disease, and STEMI or NSTEMI, were selected for multivariate analysis in our study. Multivariate logistic regression analysis revealed that older age (odds ratio [OR]: 1.05, 95% CI: 1.03–1.07, *p* < 0.001), higher Killip scores (OR: 3.77, 95% CI: 2.32–6.11, *p* < 0.001), higher peak troponin I levels (OR: 1.02, 95% CI: 1.01–1.03, *p* = 0.027), and STEMI (OR: 2.39, 95% CI: 1.43–3.98, *p* < 0.001) were associated with higher in-hospital MACE rates. By contrast, loading of ticagrelor in the ED (OR: 0.51, 95% CI: 0.31–0.85, *p* = 0.01) and higher systolic blood pressure (OR: 0.99, 95% CI: 0.98–0.99, *p* = 0.001) were associated with lower in-hospital MACE rates (Table 3).

After 1:1 matching with age, sex, prior coronary artery disease, prior cerebrovascular disease, chronic kidney disease, Killip classification, and STEMI or NSTEMI, 588 patients in each group were selected for further analysis. The demographic characteristics of the two groups were compared, as shown in Table 4. The C-statistic of PSM was 0.69, suggesting a moderate predictive performance of the matching process.

The conditional logistic regression for the matched cohort revealed a lower risk of in-hospital MACE among the ticagrelor group than the clopidogrel group (OR 0.42, 95% CI 0.21–0.85, *p* = 0.016) (Table 5).

## 4. Discussion

In this study, we found that patients with AMI who had an older age, higher Killip scores, higher peak troponin I levels, lower systolic blood pressure, and STEMI had higher in-hospital MACE rates. In addition, patients with AMI who received DAPT with ticagrelor and aspirin in the ED before PCI had lower in-hospital MACE rates than those who received DAPT with clopidogrel and aspirin.

Current guidelines recommend 1-year DAPT with ticagrelor or clopidogrel plus aspirin for patients with ACS, as well as loading doses of P2Y_12_ inhibitors and aspirin before or during primary PCI [13,14]. In the PLATO trial, compared with clopidogrel, treatment with ticagrelor significantly reduced the rate of death from vascular causes, MI, or stroke in patients with ACS [7]. Several studies comparing the long-term benefits and MACEs associated with ticagrelor versus clopidogrel [15,16,17,18,19] have reported significant reductions in the risk of MACEs following the administration of ticagrelor with aspirin. However, few studies have focused on the loading of DAPT in the ED for patients with AMI before PCI. Moreover, few studies have compared P2Y_12_ receptor antagonists for patients with AMI in the ED. Our study mainly focused on the short-term risk of MACEs associated with different DAPT regimens initiated immediately after the diagnosis of AMI in the ED. Our results revealed that DAPT with ticagrelor and aspirin in the ED before PCI is associated with lower in-hospital MACE rates.

Ticagrelor led to a consistent reduction in cardiovascular events and death, MI, and stent thrombosis and improved survival without increasing major bleeding; however, it increased the incidence rate of stroke [20]. Ticagrelor is the first reversibly binding oral P2Y_12_ receptor antagonist. Unlike clopidogrel, ticagrelor is a reversible, nonthienopyridine P2Y_12_ receptor antagonist that does not require metabolic conversion to active drugs [21]. The pharmacological and clinical profiles of ticagrelor suggest that it can provide a high and consistent level of antithrombotic protection without a proportional increase in the risk of bleeding. Moreover, it can likely offer a more rapid offset of effects than the existing thienopyridine P2Y_12_ inhibitors [22]. This may explain why the loading of ticagrelor in the ED was associated with a reduced risk of MACEs.

In the TREAT trial, compared with clopidogrel, the administration of ticagrelor after fibrinolytic therapy did not significantly reduce the frequency of cardiovascular events [23]. However, the TREAT trial only included patients with STEMI who were aged ≤75 years. In our study, all patients with AMI, including those with STEMI and NSTEMI, were enrolled. Patients with AMI who were aged ≥75 years were also enrolled. The real-world data of our study revealed a reduced risk of in-hospital MACEs in the ticagrelor group. The different patient characteristics and outcome settings may explain the different outcomes in our study. The patients in the clopidogrel group were older and had more comorbidities, such as hypertension, diabetes mellitus, coronary artery disease, cerebrovascular events, and chronic kidney disease, in addition to higher Killip scores; however, after multivariate logistic regression analysis, the differences between ticagrelor and clopidogrel were found to be an independent prognostic factor for in-hospital MACEs.

Our study has some limitations. First, as this was a single-center study, the generalizability of our findings is limited. The use of clopidogrel or ticagrelor was decided based on the physician’s clinical judgment. Given the retrospective study design, the risk of selection bias could not be avoided in this study. However, we performed propensity score matching to minimize the possible influence of selection bias. Both multivariate adjusted logistic regression and conditional logistic regression analysis revealed similar results. The C-statistic of PSM in this study was determined to be 0.69, indicating a moderate predictive performance of the matching process. It is worth noting that during the study period, there were divergent guidelines for the management of AMI between different organizations. While the American College of Cardiology and American Heart Association recommended ticagrelor over clopidogrel for AMI patients [14], guidelines in Taiwan advised the preference of ticagrelor until they were updated in 2018 [24]. Given the potential variation in the choice of DAPT regimen for AMI patients during the period of guideline discrepancy, the C-statistic might have displayed some relative unsatisfactory aspects despite the matching process aiming to account for various clinical confounders. Third, our following time was end to the patient’s discharge from the hospital. The long-term prognostic effect on cardiovascular death or hospitalization for heart failure could not be estimated in our study. In addition, although we excluded a small percentage of patients because of incomplete data, it is difficult to predict whether including these patients would have affected the statistical results. Finally, we did not record any bleeding event and did not assess adverse events associated with different P2Y_12_ receptor antagonists. Additional studies are warranted to provide information on potential improvement in outcomes in patients with AMI.

## 5. Conclusions

Patients with AMI treated with ticagrelor and aspirin have lower in-hospital MACE rates than those treated with clopidogrel and aspirin in the ED before PCI.

## Figures and Tables

**Figure 1 healthcare-11-02246-f001:**
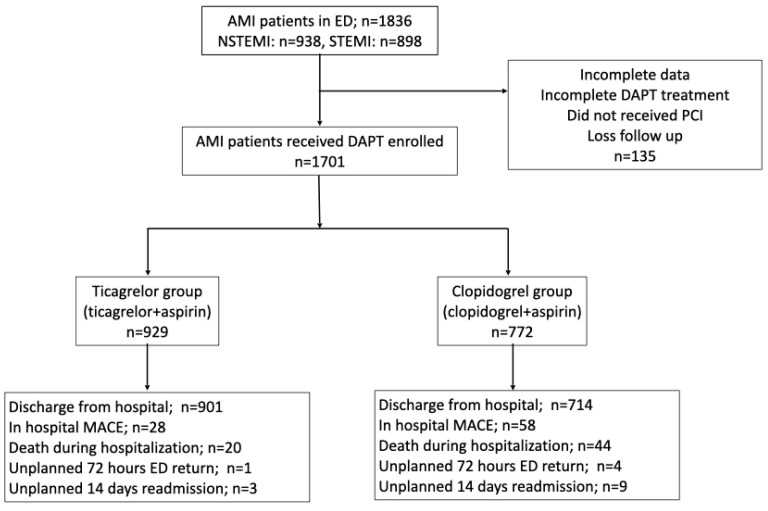
Flowchart of patient enrollment. AMI, acute myocardial infarction; ED, emergency department; NSTEMI, non-ST-segment elevation myocardial infarction; STEMI, ST-segment elevation myocardial infarction; DAPT, dual antiplatelet therapy; PCI, percutaneous coronary intervention; MACE, major adverse cardiovascular event.

**Table 1 healthcare-11-02246-t001:** Patient characteristics.

Variables	Drug		*p*-Value
	Clopidogrel (*n* = 772)	Ticagrelor (*n* = 929)	
Age, mean ± SD	65.63 ± 13.49	59.37 ± 13.08	<0.001 ^a^
Sex (%):			<0.001 ^b^
Male	555 (71.89)	764 (82.24)	
Female	217 (28.11)	165 (17.76)	
Clinical history (%):			
Smoking	369 (47.80)	537 (57.80)	<0.001 ^b^
Hypertension	483 (62.56)	509 (54.79)	0.001 ^b^
Diabetes mellitus	337 (43.65)	295 (31.75)	<0.001 ^b^
Coronary artery disease	228 (29.53)	156 (16.79)	<0.001 ^b^
Cerebrovascular disease	61 (7.90)	27 (2.91)	<0.001 ^b^
Chronic kidney disease	174 (22.54)	87 (9.36)	<0.001 ^b^
Hyperlipidemia	41 (5.31)	85 (9.15)	0.002 ^b^
Presentation features:			
Killip class			
Level 2–Level 4	182 (23.58)	145 (15.61)	<0.001 ^b^
Systolic BP (SBP)	136.5 ± 39.21	135.2 ± 32.56	0.472 ^a^
Diastolic BP (DBP)	83.55 ± 26.71	86.06 ± 22.75	0.039 ^a^
Heart rate, bpm	85.96 ± 27.09	83.27 ± 22.56	0.028 ^a^
Peak troponin I, ng/mL	5.19 ± 11.53	3.60 ± 9.36	0.002 ^a^
GFR	54.10 ± 34.35	67.52 ± 27.88	<0.001 ^a^
Creatinine	2.64 ± 3.10	1.67 ± 3.88	<0.001 ^a^
LVEF	49.99 ± 12.17	50.36 ± 11.23	0.535 ^a^
Angiographic findings:			
Left main disease *	54 (6.99)	56 (6.03)	0.419 ^b^
No. of disease vessels *			0.107 ^b^
1	355 (45.98)	445 (47.90)	
2	250 (32.38)	293 (31.54)	
3	111 (14.38)	148 (15.93)	
No. of disease vessels *			0.374 ^b^
<3	661 (85.62)	781 (84.07)	
≥3	111 (14.38)	148 (15.93)	
Type:			<0.001 ^b^
STEMI	292 (37.82)	543 (58.45)	
NSTEMI	480 (62.18)	386 (41.55)	

Data are presented as the means ± standard deviations for continuous variables and numbers (percentages) for categorical variables. ^a^ Two-sample *t*-test. ^b^ Chi-squared test. * Significant disease refers to >75% stenosis of a coronary artery, except in left main coronary artery disease (stenosis > 50%). BP: blood pressure; GFR: glomerular filtration rate; LVEF: left ventricular ejection fraction; STEMI: ST-elevation myocardial infarction; NSTEMI: non-ST-elevation myocardial infarction.

**Table 2 healthcare-11-02246-t002:** Patient outcomes in each group.

Variables	Drug		*p*-Value
	Clopidogrel (*n* = 772)	Ticagrelor (*n* = 929)	
In-hospital MACE (%):			<0.001 ^a^
No	714 (92.49)	901 (96.99)	
Stroke	10 (1.30)	5 (0.54)	
Recurrent MI	1 (0.13)	2 (0.22)	
Unplanned repeat PCI	3 (0.39)	1 (0.11)	
Death	44 (5.70)	20 (2.15)	
72 h ED return (%):			0.096 ^a^
No	768 (99.48)	925 (99.57)	
Planned	0 (0.00)	3 (0.32)	
Unplanned	4 (0.52)	1 (0.11)	
14 days readmission (%):			0.096 ^a^
No	755 (97.80)	919 (98.92)	
Planned	8 (1.04)	7 (0.75)	
Unplanned	9 (1.17)	3 (0.32)	
PCI performed within 30 days	106 (13.73)	160 (17.22)	0.048 ^a^
CABG performed within 30 days	5 (0.65)	10 (1.08)	0.346 ^a^

^a^ Chi-squared test. MACE: major adverse cardiovascular event; MI: myocardial infarction; PCI: percutaneous coronary intervention; ED: emergency department; CABG: coronary artery bypass graft.

**Table 3 healthcare-11-02246-t003:** Univariate and multivariate analyses of in-hospital MACE rates.

Parameters	Univariate		Multivariate	
	OR (95% CI)	*p*-Value	OR (95% CI)	*p*-Value
Age	1.06 (1.04–1.08)	<0.001	1.05 (1.03–1.07)	<0.001
Sex, Male	0.65 (0.41–1.05)	0.078	1.16 (0.66–2.04)	0.604
Drug:				
clopidogrel	Reference	-	Reference	-
ticagrelor	0.38 (0.24–0.61)	<0.001	0.53 (0.32–0.88)	0.013
Clinical history:				
Smoking	0.53 (0.34–0.83)	0.005	0.86 (0.50–1.46)	0.567
Hypertension	1.60 (1.01–2.55)	0.048	1.12 (0.66–1.89)	0.677
Diabetes mellitus	1.58 (1.02–2.44)	0.039	1.08 (0.66–1.77)	0.769
Coronary artery disease	1.27 (0.78–2.07)	0.343		
Cerebrovascular disease	3.28 (1.71–6.30)	<0.001	1.75 (0.84–3.64)	0.133
Chronic kidney disease	1.18 (0.66–2.09)	0.579		
Hyperlipidemia	0.76 (0.30–1.92)	0.564		
Presentation features:				
Killip class				
Level 1	Reference	-	Reference	-
Level 2–Level 4	6.37 (4.08–9.95)	<0.001	3.77 (2.32–6.11)	<0.001
Systolic BP (SBP)	0.99 (0.98–0.99)	<0.001	0.99 (0.98–0.99)	0.001
Diastolic BP (DBP)	0.98 (0.97–0.99)	<0.001		
Heart rate	1.00 (0.99–1.01)	0.365		
Peak troponin I	1.02 (1.01–1.04)	0.001	1.02 (1.01–1.03)	0.027
Angiographic findings:				
Left main disease *	2.25 (1.16–4.37)	0.017	1.82 (0.87–3.82)	0.110
No. of disease vessels *				
<3	Reference	-		
≥3	1.63(0.96–2.75)	0.071		
Type:				
NSTEMI	Reference	-	Reference	-
STEMI	1.71 (1.10–2.67)	0.018	2.39 (1.43–3.98)	<0.001

* Significant disease refers to >75% stenosis of a coronary artery, except in left main coronary artery disease (stenosis > 50%). NSTEMI: non-ST-elevation myocardial infarction; STEMI: ST-elevation myocardial infarction.

**Table 4 healthcare-11-02246-t004:** After 1:1 propensity score matching of patient characteristics.

Variables	Drug		SMD
	Clopidogrel (*n* = 588)	Ticagrelor (*n* = 588)	
Age, mean ± SD	63.63 ± 13.52	63.06 ± 12.72	−0.043
Sex (%):			0.048
Male	445 (75.68)	457 (77.72)	
Female	143 (24.32)	131 (22.28)	
Clinical history (%):			
Smoking	312 (53.06)	313 (53.23)	0.003
Hypertension	343 (58.33)	345 (58.67)	0.006
Diabetes mellitus	226 (38.44)	204 (34.69)	−0.077
Coronary artery disease	128 (21.77)	124 (21.09)	−0.016
Cerebrovascular disease	24 (4.08)	24 (4.08)	0.000
Chronic kidney disease	78 (13.27)	80 (13.61)	0.009
Hyperlipidemia	32 (5.44)	52 (8.84)	0.132
Presentation features:			
Killip class			
Level 2–Level 4	106 (18.03)	107 (18.20)	0.004
Systolic BP (SBP)	135.9 ± 40.60	135.4 ± 32.68	−0.012
Diastolic BP (DBP)	84.80 ± 27.74	84.91 ± 22.46	0.004
Heart rate, bpm	84.65 ± 27.51	84.71 ± 22.98	0.002
Peak troponin I, ng/mL	4.74 ± 11.15	4.33 ± 10.27	−0.038
GFR	61.64 ± 32.38	62.54 ± 29.13	0.029
Creatinine	2.07 ± 2.67	1.94 ± 4.77	−0.033
LVEF	50.86 ± 12.22	50.74 ± 11.06	−0.009
Angiographic findings:			
Left main disease *	27 (4.59)	42 (7.14)	0.108
No. of disease vessels *			0.117
1	277 (47.11)	273 (46.43)	
2	177 (30.10)	175 (29.76)	
3	89 (15.14)	108 (18.37)	
No. of disease vessels *			0.086
<3	499 (84.86)	480 (81.63)	
≥3	89 (15.14)	108 (18.37)	
Type:			0.044
STEMI	268 (45.58)	255 (43.37)	
NSTEMI	320 (54.42)	333 (56.63)	

Data are presented as the means ± standard deviations for continuous variables and numbers (percentages) for categorical variables. SMD: standardized mean difference. * Significant disease refers to >75% stenosis of a coronary artery, except in left main coronary artery disease (stenosis > 50%). BP: blood pressure; GFR: glomerular filtration rate; LVEF: left ventricular ejection fraction; STEMI: ST-elevation myocardial infarction; NSTEMI: non-ST-elevation myocardial infarction.

**Table 5 healthcare-11-02246-t005:** After 1:1 propensity score matching and univariate and multivariate analyses for in-hospital MACE rates.

Drug	Univariate		Multivariate	
	OR (95% CI)	*p*-Value	OR (95% CI)	*p*-Value
Clopidogrel	Reference	-	Reference	-
Ticagrelor	0.46 (0.27–0.80)	0.005	0.42 (0.21–0.85)	0.016

OR: odds ratio; CI: confidence interval.

## Data Availability

The datasets generated and analyzed in this study are not publicly available due to the nondisclosure agreement of the Institutional Review Board. The datasets are available from the corresponding author upon reasonable request.

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
