# Peer review of "Comparison of In-Hospital Major Adverse Cardiovascular Events in Patients with Acute Myocardial Infarction Treated with Ticagrelor or Clopidogrel in the Emergency Department: A Propensity Score Matched Retrospective Cohort Study"

_healthcare, 2023, doi:10.3390/healthcare11162246_

Round 1
Reviewer 1 Report
Dual antiplatelet therapy is recommended in patients with acute coronary syndrome as it reduces the rate of major adverse cardiovascular events. If initially, clopidogrel was the indication in association with aspirin, Further reductions in events have been demonstrated with the replacement of clopidogrel by ticagrelor or prasugrel. But this seems to be still a matter of debate, because of the increased risk of bleeding in patients under ticagrelor demonstrated by some authors, or because of similar bleeding and ischemic outcomes comparing ticagrelor with clopidogrel. Some other studies did not show a statistically significant difference in major adverse cardiovascular events or at least showed less bleeding in patients treated with ticagrelor than in those with clopidogrel.
The guidelines currently recommend ticagrelor over clopidogrel and this population-based study with an interesting and significant sample of patients, seems to confirm a better outcome with ticagrelor, even if there may be a little bias in the selection of the patients, with patients with higher baseline risk more prone to experience bleeding.
Needs some minor editing and a review of the abbreviations
Author Response
Dual antiplatelet therapy is recommended in patients with acute coronary syndrome as it reduces the rate of major adverse cardiovascular events. If initially, clopidogrel was the indication in association with aspirin, Further reductions in events have been demonstrated with the replacement of clopidogrel by ticagrelor or prasugrel. But this seems to be still a matter of debate, because of the increased risk of bleeding in patients under ticagrelor demonstrated by some authors, or because of similar bleeding and ischemic outcomes comparing ticagrelor with clopidogrel. Some other studies did not show a statistically significant difference in major adverse cardiovascular events or at least showed less bleeding in patients treated with ticagrelor than in those with clopidogrel.
The guidelines currently recommend ticagrelor over clopidogrel and this population-based study with an interesting and significant sample of patients, seems to confirm a better outcome with ticagrelor, even if there may be a little bias in the selection of the patients, with patients with higher baseline risk more prone to experience bleeding.
Needs some minor editing and a review of the abbreviations
REPLY: Thanks for reviewer’s recommendation and we had reviewed the abbreviations.
Reviewer 2 Report
This study is to investigate the in-hospital prognosis of patients with acute myocardial infarction treated with ticagrel and Clopidogrel. However, some issues may limit this work.
1.The title of this manuscript needs to be more precise, indicating the observation of adverse events during hospitalization.
2. After PSM, it is recommended to use the standardized difference method instead of the P-value for the balance test indicators between the two groups.
3. PSM setting conditions need to be supplemented in statistical analysis, including but not limited to: matching method, fixed caliper width. In addition, the evaluation method for the stability of the model established after PSM needs to be supplemented.
4. Although the author used PSM, there is still a serious imbalance in the matched inter group data, especially in indicators such as Killip grading and chronic kidney disease that have a serious impact on the occurrence of MACEs. It is recommended that the author redo PSM, and the standardized difference method is a good choice.
5. After PSM, the conditional logistic model may be relatively more suitable than the non conditional logistic model.
Author Response
This study is to investigate the in-hospital prognosis of patients with acute myocardial infarction treated with ticagrelor and Clopidogrel. However, some issues may limit this work.
1.The title of this manuscript needs to be more precise, indicating the observation of adverse events during hospitalization.
REPLY: Thanks for the reviewer’s suggestion, we decided to change the title “Comparison of In-Hospital Major Adverse Cardiovascular Event Between Patients with Acute Myocardial Infarction Treated with Ticagrelor and Clopidogrel in the Emergency Department: A Propensity Score Matched Retrospective Cohort Study”
- After PSM, it is recommended to use the standardized difference method instead of the P-value for the balance test indicators between the two groups.
REPLY: Thanks again for this suggestion, we used standardized mean difference instead of the P-value for the balance test indicators between the 2 groups and was shown in table 4 (Line 222, Page 7)
- PSM setting conditions need to be supplemented in statistical analysis, including but not limited to: matching method, fixed caliper width. In addition, the evaluation method for the stability of the model established after PSM needs to be supplemented.
REPLY: Appreciated for this important comment. We had further described the PSM setting condition in the manuscript. In this study, PSM was performed using the nearest neighbor method, with a caliper width set at 0.1. After PSM, the standardized mean difference was utilized to assess the balance between the clopidogrel and ticagrelor groups, and the C-statistic was calculated to evaluate the predictive performance of the propensity score model. Subsequently, a conditional logistic regression analysis was conducted, adjusting for the same covariates, to identify the effects of ticagrelor and clopidogrel on the reduction of in-hospital MACE. (line 140, page 3 to line 146, page 4). The result of C-statistic was showed in Line 220, page 7 and the discussion of the C-statistic result was prescribed at Line 284 to 288, page 10.
- Although the author used PSM, there is still a serious imbalance in the matched inter group data, especially in indicators such as Killip grading and chronic kidney disease that have a serious impact on the occurrence of MACEs. It is recommended that the author redo PSM, and the standardized difference method is a good choice.
REPLY: We appreciated this important suggestion and we had redone PSM according to the reviewer’s suggestion. Factors that might influence the choice of ticagrelor or clopidogrel, including age, sex, prior coronary artery disease, prior cerebrovascular disease, chronic kidney disease, Killip classification, and STEMI or NSTEMI, were selected for PSM (the text was prescribed in Line 138, page 3). Chronic kidney disease and Killip classification were included for the redo PSM and the result were showed in table 4 and table 5, which showed minor difference in multivariate result after PSM (shown in line 28, 29, page 1, line 229 to 231, page 8 and Table 5).
- After PSM, the conditional logistic model may be relatively more suitable than the non conditional logistic model.
REPLY: Thanks for suggestion, we had performed the conditional logistic model initially and the result was showed in table 5.
Reviewer 3 Report
1. Please explain why the loading dose of clopidogrel was 300mg and not 600mg, at least in those with STEMI.
2. In the section 2.3. in the first sentence correct the term "rupture PCI"
3. Did you performed multiplate analysis and could taht affect the results?
4. Why the bleeding complications were not recorded?
Minor changes should be done.
Author Response
- Please explain why the loading dose of clopidogrel was 300mg and not 600mg, at least in those with STEMI.
REPLY: The recommendation of loading dose of clopidogrel was 300-600mg in STEMI patients undergoing primary PCI of clopidogrel in 2020 Focused Update of the 2012 guidelines of the Taiwan Society of Cardiology for the management of STEMI. The consensus between ED physicians and cardiologists in most Taiwan hospitals and our hospital was loading dose of 300mg clopidogrel. Because of the character of average low body weights of Asians and according to the CURRENT-OASIS 7 trial, which showed clopidogrel 600mg loading dose did not reduce ischemic events but lead to higher major bleeding risk comparing with 300mg loading dose in patients with ACS.
- In the section 2.3. in the first sentence correct the term "rupture PCI"
REPLY: Appreciated for the correction, rupture PCI will be corrected to “unplanned repeat PCI”,and delete unnecessary word “or death”. (Line 121, page 3 and table 2.)
- Did you performed multiplate analysis and could taht affect the results?
REPLY: Excuse me, did the reviewer mean “Multiple analysis”? The primary outcome of our study is in-hospital MACE, which belongs to category result, so we conducted multivariate adjusted logistic regression to evaluate the difference risk of MACE between Ticagrelor and Clopidogrel groups.
- Why the bleeding complications were not recorded?
REPLY: In our point of view, previous studies (the PLATO study) had showed no significant difference in the rates of major bleeding was found between the ticagrelor and clopidogrel group, we thought the effected of bleeding tendency in each P2Y12 inhibitor is beyond debate. When we designed this retrospective study, we decided to investigate the outcome of MACE and decided not to collect the bleeding events data due to previous PLATO study results.